# Unraveling the causal association between leukocyte telomere length and infertility: A two-sample Mendelian randomization study

Gaole An[1�], Xingnan Zhao[2☉], Chenghui Zhao[3,4]*

1 Information Department, Bethune International Peace Hospital, Shijiazhuang, Hebei Province, China, 2 Department of Obstetrics and Gynecology, Bethune International Peace Hospital, Shijiazhuang, Hebei Province, China, 3 Key Laboratory of Biomedical Engineering and Translational Medicine, Ministry of Industry and Information Technology, Beijing, China, 4 Research Center for Biomedical Engineering, Medical Innovation & Research Division, Chinese PLA General Hospital, Beijing, China

☉ These authors contributed equally to this work.
* zhaochenghui@plagh.org

**Data Availability Statement:** The summary statistics of GWAS dataset for metabolic traits can be accessed from IEU open GWAS project (https://gwas.mrcieu.ac.uk/datasets/) website under the accession ID provided in S1 Table.

## Abstract

Infertility is a significant challenge in modern society, and observed studies have reported the association between telomere length and infertility. Whether this relationship is causal remains controversial. We employed two-sample mendelian randomization (MR) to investigate the causal relationship between leukocyte telomere length (LTL) and major causes of infertility, including male and female infertility, sperm abnormalities, and endometriosis. MR analyses were mainly performed using the inverse variance weighted (IVW) method and complemented with other MR methods. Our findings demonstrate a causal association between LTL and endometriosis (OR1.304, 95% CI (1.122,1.517), p = 0.001), suggesting its potential as a biomarker for this condition. However, we did not observe a significant causal relationship between LTL and other infertility causes. Our study presents compelling evidence on the relationship between LTL and endometriosis. Meanwhile, our study demonstrates that there is no causal relationship between LTL and infertility. This research contributes to the field by shedding light on the importance of LTL in the early diagnosis and intervention of endometriosis.

## Introduction

Telomeres, the protective caps at the ends of eukaryotic linear chromosomes [1], play a crucial role in maintaining genomic stability and cellular health [2]. Shortened telomeres have been associated with various age-related diseases and conditions, including cancer, cardiovascular disease, and neurodegenerative disorders [3]. In recent years, there has been growing interest in exploring the potential link between telomere length and infertility, a condition that affects a significant proportion of the population.

With the rapid modernization of society, infertility has emerged as a formidable hurdle for individuals and couples worldwide, with a prevalence ranging from 10% to 15% [4]. It encompasses a range of conditions, including male and female infertility, sperm abnormalities, and

**Funding:** The author(s) received no specific funding for this work.

**Competing interests:** The authors have declared that no competing interests exist.

endometriosis. Understanding the underlying mechanisms and potential biomarkers for infertility is crucial for developing effective diagnostic tools and therapeutic interventions. Observed studies have found the relationship between leukocyte telomere length (LTL) and infertility. Shorter telomere length was associated with female infertility factors, such as polycystic ovary syndrome (PCOS), ovarian insufficiency and tubal factor in oocyte granulosa cells, endometrial tissue and leukocytes [5]. And shorter LTL was also associated with greater odds of endometriosis [6]. As for male infertility, the mean of LTL and sperm telomere length (STL) were significantly shorter in infertile men compared with fertile individuals [7]. However, conflicting results regarding the association between leukocyte telomere length and infertility have been observed in numerous RCT studies [5], and the causal relationship and underlying mechanisms are still to be elucidated.

Mendelian randomization (MR), a method that utilizes genetic variants as instrumental variables, offers a unique opportunity to investigate causal relationships between exposures and outcomes [8]. Unlike traditional observational studies, which are prone to confounding and reverse causality, mendelian randomization leverages genetic variants that are randomly assigned at conception and are not influenced by confounders or disease status. This approach allows researchers to overcome limitations of observational studies and provide more robust evidence for causal associations [9]. By leveraging genetic variants that are associated with telomere length, Mendelian randomization can provide insights into the causal association between telomere length and infertility.

In this study, we aim to utilize two-sample Mendelian randomization to explore the causal relationship between leukocyte telomere length and major causes of infertility. Specifically, we will investigate the potential causal association between telomere length and male and female infertility, sperm abnormalities, and endometriosis. By utilizing large-scale genetic data and statistical approaches, we aim to provide robust evidence regarding the role of telomere length in infertility.

## Materials and methods

### Study design

In our research, we employed a two-sample mendelian randomization approach to meticulously examine the causal impacts of LTL on various aspects of infertility, encompassing male and female infertility, sperm abnormalities, and endometriosis. The MR study based on three fundamental assumptions: (1) the instrumental variables (single nucleotide polymorphisms or SNPs) exhibit robust associations with the exposures under scrutiny; (2) the instrumental variables are not linked to any confounding factors; (3) the instrumental variables exclusively influence the outcome through the exposure, without exerting any direct effect on the outcome itself. The schematic representation of our study design is visually depicted in Fig 1.

### Data source

**Exposure data.** The genetic association data for LTL were obtained from a genome-wide association study (GWAS) involving 472,174 European participants [10], adjusting for age, sex, and the first ten principal components (PCs). These participants were aged 40–69 years and had an equal distribution of males (45.8%) and females (54.2%). LTL measurements were determined using a quantitative polymerase chain reaction (qPCR) assay. To mitigate any potential biases stemming from body mass index (BMI) and smoking, we procured comprehensive BMI GWAS summary data from the UK Biobank and smoking GWAS sunmmary data from the GWAS & Sequencing Consortium of Alcohol and Nicotine use (GSCAN) [11].

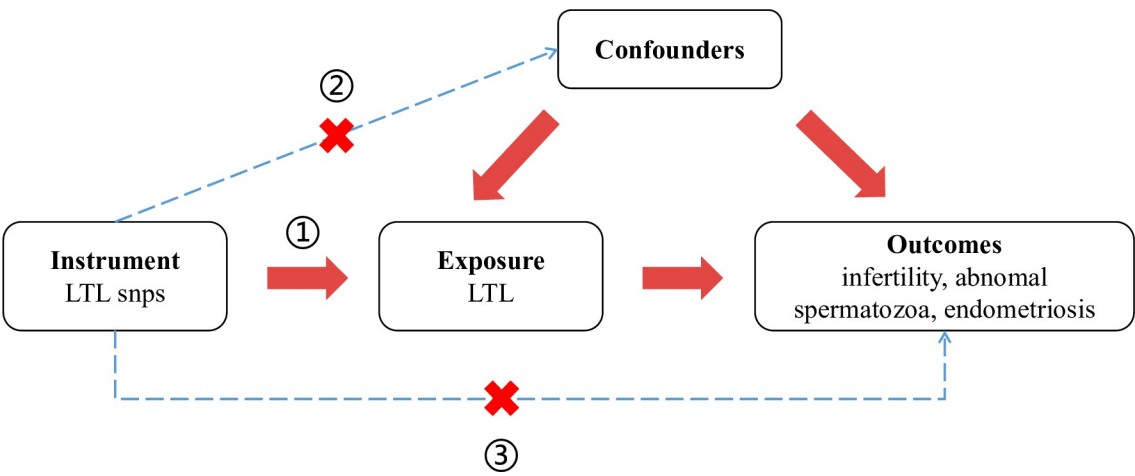

**Fig 1. Schematic illustration of the study design.** LTL leukocyte telomere length, snps single nucleotide polymorphisms.

Leveraging these datasets, we conducted multivariable Mendelian Randomization (MVMR) to ensure a robust analysis.

**Outcome data.** In this study, we investigated the outcomes of male infertility, female infertility, sperm abnormalities, and endometriosis. We utilized the Ieu Open GWAS Project database (https://gwas.mrcieu.ac.uk/) to identify the most suitable GWAS summary data that align with our research objectives. Finally, we obtained summary data on infertility from the FinnGen Consortium R5 release and the UK Biobank. S1 Table provides detailed information about the specific outcomes examined in our analysis. The diagnostic criteria for infertility-related diseases were based on the International Classification of Diseases (ICD) codes, including ICD-9 and ICD-10. Female infertility was further categorized into different subtypes, including Female infertility, Female infertility associated with anovulation, Female infertility of cervigal, vaginal, other or unspecified origin, and Female infertility of tubal origin. In addition, we investigated endometriosis in various organs, such as the fallopian tube, intestine, ovary, pelvic peritoneum, rectovaginal septum and vagina, uterus, and unspecified locations.

## The IV selection

Instrument variables associated with LTL, BMI and smoking in GWAS datasets were selected using a stringent set of inclusion criteria. Initially, SNPs with genome-wide significance ($p \leq 5 \times 10^{-8}$) were chosen. Subsequently, we performed clumping of SNPs by excluding variants in linkage disequilibrium (LD, $R^2 > 0.001$ and within 10,000 kb). To ensure consistency, all selected SNPs were harmonized to correspond to the same allele for effect estimation. The strength of the instrumental variables (IVs) was assessed using the F statistic, with a threshold of less than 10 defining weak instruments that were subsequently excluded [12]. Furthermore, we eliminated palindromic SNPs that could introduce uncertainty in determining the effect allele in the exposure GWASs. After this rigorous screening process, the remaining SNPs were deemed eligible instrumental variables. S2 Table provides comprehensive information on all instrumental variables used in this study, and S3 Table gives summary information of these instrument variables.

## Statistics analysis

We utilized various statistical techniques, such as inverse variance weighting (IVW), the weighted median (WM), MR-Egger, the weighted model, and the simple model, to assess the

causal association between exposure (LTL) and outcome (infertility). The IVW method was chosen as the primary statistical analysis approach, with the random effects model used in the presence of heterogeneity [13].

Various sensitivity analyses were conducted to assess the robustness of the findings. These included tests for heterogeneity, pleiotropy, leave-one-out analyses, and the Mendelian randomization pleiotropy residual sum and outlier (MR-PRESSO) method [14]. Heterogeneity was evaluated using Cochran's Q statistic, with a significance level of $p < 0.05$ indicating the presence of heterogeneity [15]. Directional pleiotropy was assessed using the MR-Egger intercept analysis, with a p-value greater than 0.05 suggesting the absence of pleiotropy [16]. The MR-PRESSO method was employed to identify and remove outliers that may have influenced the results. Additionally, the leave-one-out test involved systematically removing single nucleotide polymorphisms (SNPs) one by one and assessing the stability of the results. Stable and reliable causal relationships were indicated if the remaining results did not exhibit significant changes. To address potential confounding effects stemming from BMI) and smoking, we utilized the MVMR to investigate the influence of BMI, leukocyte telomere length (LTL), and smoking on the occurrence of infertility, sperm abnormalities, and endometriosis.

The results were reported as odds ratios (ORs) with corresponding 95% confidence intervals. Two-sided p-values were used, and statistical significance was determined at $p < 0.05$. All statistical analyses were performed using the "Two-Sample MR" packages in R software (version 4.3.1).

## Result

### LTL and male infertility

We conducted a comprehensive analysis using the male infertility and sperm abnormalities datasets from FinnGen to evaluate the causal association between leukocyte telomere length (LTL) and the occurrence of male infertility and sperm abnormalities. The findings from Table 1 indicate that no causal relationships were observed between LTL and male infertility (OR 1.269, 95%CI (0.838, 1.921), p = 0.261). Similarly, there were no causal associations found between LTL and sperm abnormalities (OR 0.918, 95%CI (0.640, 1.317), p = 0.643). The analysis did not uncover any heterogeneity or pleiotropy.

### LTL and female infertility

To investigate the potential causal relationship between leukocyte telomere length (LTL) and female infertility, we conducted a rigorous analysis utilizing the comprehensive collection of five datasets obtained from FinnGen. The analysis revealed no causal relationships between LTL and female infertility (OR 1.108, 95%CI (0.962, 1.277), p = 0.155). Consistently, LTL did not have any causal effects on female infertility caused by anovulation (OR 0.932, 95% CI (0.667, 1.303), p = 0.682), endometriosis (OR 1.174, 95%CI (0.891, 1.547), p = 0.255), tubal factors (OR 1.154, 95%CI (0.760, 1.753), p = 0.502), or other factors related to cervigal, vaginal,

**Table 1. Results of MR analysis between LTL and male infertility.**

| Outcomes | OR | 95%CI | p | $p_{pleiotropy}$ | $p_{heterogeneity}$ | $p_{mr-presso}$ | $p_{MVMR}$ |
|---|---|---|---|---|---|---|---|
| Male infertility | 1.269 | (0.838,1.921) | 0.261 | 0.825 | 0.97 | 0.974 | 0.286 |
| Abnormal spermatozoa | 0.918 | (0.640,1.317) | 0.643 | 0.842 | 0.771 | 0.781 | 0.381 |

MR, Mendelian randomization; LTL, leukocyte telomere length; OR, odds ratio; CI, confidence interval; p, p value; MVMR, multivariable Mendelian randomization.

**Table 2. Results of MR analysis between LTL and female infertility.**

| Outcomes | OR | 95%CI | p | $p_{pleiotropy}$ | $p_{heterogeneity}$ | $p_{mr-presso}$ | $p_{MVMR}$ |
|---|---|---|---|---|---|---|---|
| Female infertility | 1.108 | (0.962,1.277) | 0.155 | 0.199 | 0.547 | 0.557 | 0.132 |
| Female infertility, associated with anovulation | 0.932 | (0.667,1.303) | 0.682 | 0.159 | 0.623 | 0.630 | 0.679 |
| Female infertility, cervigal, vaginal, other or unspecified origin | 1.086 | (0.937,1.259) | 0.273 | 0.164 | 0.647 | 0.667 | 0.175 |
| Female infertility, tubal origin | 1.154 | (0.760,1.753) | 0.502 | 0.774 | 0.100 | 0.086 | 0.645 |
| Endometriosis diagnosis and infertility diagnosis occurring together | 1.174 | (0.891,1.547) | 0.255 | 0.396 | 0.536 | 0.529 | 0.640 |

MR, Mendelian randomization; LTL, leukocyte telomere length; OR, odds ratio; CI, confidence interval; p, p value; MVMR, multivariable Mendelian randomization.

and other organs (OR 1.086, 95%CI (0.937, 1.259), p = 0.273) (Table 2). The analysis did not uncover any heterogeneity or pleiotropy.

## LTL and endometriosis

We utilized three datasets from FinnGen and UKbiobank to assess the causal relationships between LTL and endometriosis. In the FinnGen dataset, heterogeneity ($P_{heterogeneity}$ = 0.023) and pleiotropy ($P_{MR-PRESSO}$ = 0.019) were observed in this dataset. After removing outliers identified by MR-PRESSO, the MR analyses indicated a causal relationship between LTL and endometriosis. Specifically, the risk of endometriosis increased by 0.304 times with a one standard deviation decrease in genetically predicted LTL, as determined by the random effects IVW method (OR = 1.304, 95% CI = 1.122–1.517, p = 0.001) (Table 3). In the UKbiobank dataset, no clear evidence of heterogeneity or pleiotropy was found, leading us to employ the IVW method with a fixed effects model for causal estimation. We also identified causal relationships between LTL and a higher risk of self-reported endometriosis (OR 1.004, 95%CI (1.002, 1.005), p = 2.65E-5) and ICD10 diagnosed endometriosis (OR 1.002, 95%CI (1.000,1.003), p = 0.049). The effects of the SNPs on LTL and endometriosis are depicted in Fig 2 through scatter plots. After excluding the effects of confounding factors such as BMI and smoking, the results of MVMR showed that the causal effect of LTL on both FinnGen dataset ($P_{MVMR}$ = 0.023) and self-reported endometriosis ($P_{MVMR}$ = 0.002) remained significant.

Furthermore, we investigated the causal effects of LTL on endometriosis in different organs. The forest plot (Fig 3) revealed that the causal effects of LTL were particularly pronounced in endometriosis of the intestine (OR 3.58, 95%CI (1.68, 8.02), p = 0.002) and ovary (OR 1.37, 95%CI (1.10, 1.70), p = 0.004). Additional details regarding the results of other statistical methods, as well as the results of pleiotropy and heterogeneity testing, can be found in S4–S7 Tables. The leave-one-out analysis demonstrated consistent results with the MR studies, indicating that no single SNP significantly influenced the findings. This suggests the robustness and reliability of the MR studies (S1 Fig).

**Table 3. Results of MR analysis between LTL and endometriosis.**

| Outcomes | Consortiumm | OR | 95%CI | p | $p_{pleiotropy}$ | $p_{heterogeneity}$ | $p_{mr-presso}$ | $p_{MVMR}$ |
|---|---|---|---|---|---|---|---|---|
| Endometriosis | FinnGen | 1.304 | (1.122,1.517) | 0.001 | 0.896 | 0.023 | 0.019 | 0.023 |
| ICD10: N80.0 Endometriosis of uterus | UKbiobank | 1.002 | (1.000,1.003) | 0.049 | 0.912 | 0.269 | 0.270 | 0.124 |
| Self-reported: endometriosis | UKbiobank | 1.004 | (1.002,1.005) | 2.65E-05 | 0.870 | 0.164 | 0.155 | 0.002 |

MR, Mendelian randomization; LTL, leukocyte telomere length; OR, odds ratio; CI, confidence interval; p, p value; MVMR, multivariable Mendelian randomization.

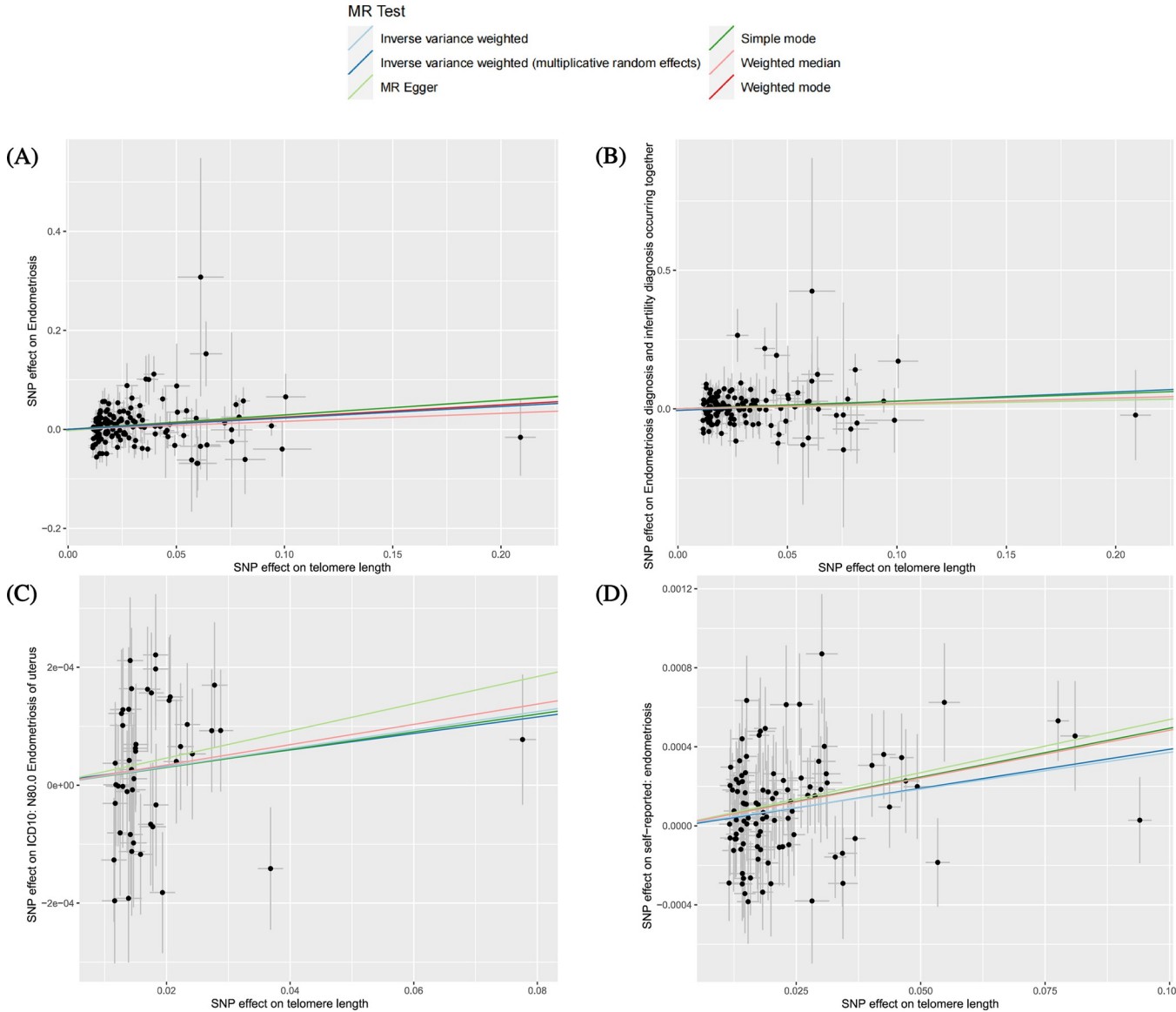

**Fig 2. Scatter plot illustrating the distribution of individual ratio estimates of LTL with endometriosis as the outcome.** (A) Endometriosis as the outcome. (B) Endometriosis and infertility occurring together as the outcome. (C) ICD10 endometriosis of uterus as the outcome. (D) Self-reported endometriosis as the outcome. Trend lines derived from five different MR methods are also included in each scatter plot to indicate cause and effect. LTL, leukocyte telomere length; MR, Mendelian Randomization; ICD, International classification of Diagnose.

## Discussion

Infertility poses a significant global challenge, and our study firstly utilized Mendelian randomization to explore the link between leukocyte telomere length and male and female infertility. Our findings reveal a noteworthy causal relationship between LTL and endometriosis, indicating that individuals with shorter telomeres are more likely to have this condition. However, our investigation did not yield evidence of a substantial causal association between LTL and infertility in general.

An insightful review has suggested that telomere length (TL) can be regarded as a molecular marker for assessing spermatogenesis and sperm quality, and may also have implications for

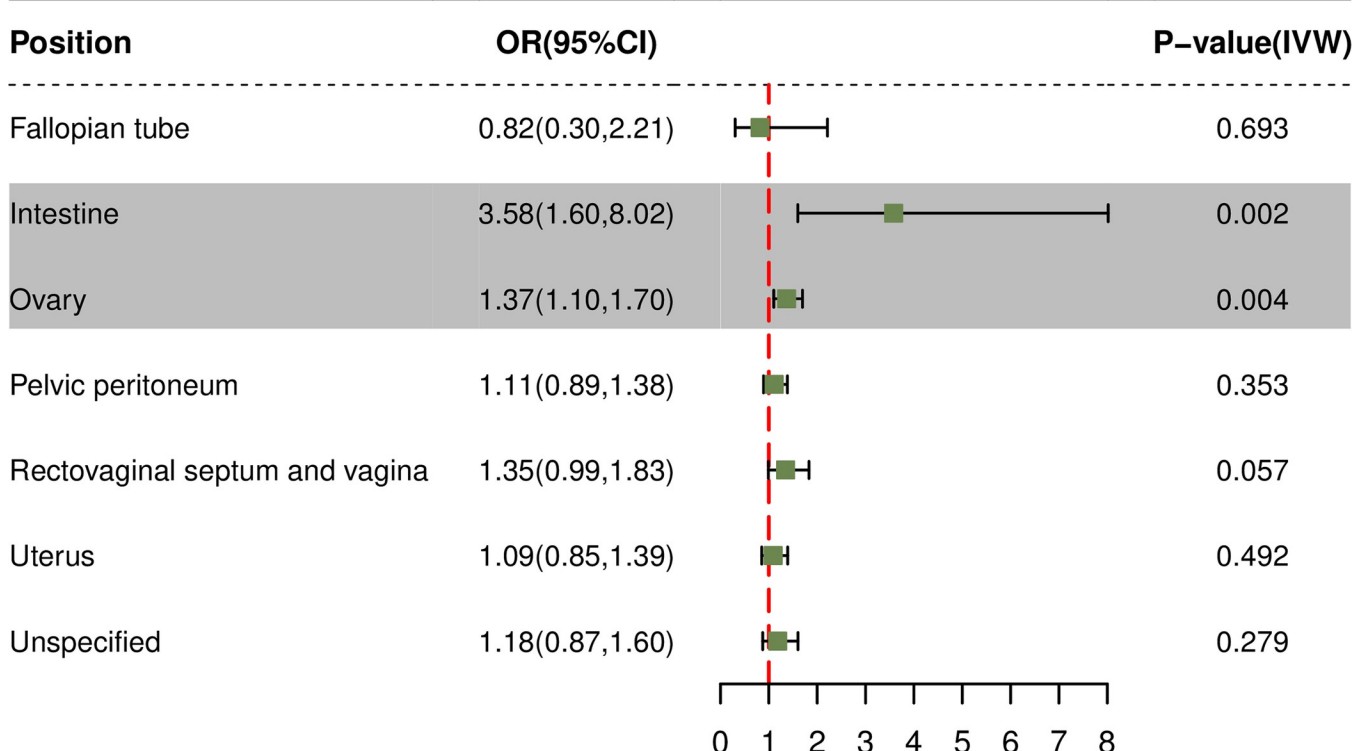

| Position | OR(95%CI) | | P-value(IVW) |
|---|---|---|---|
| Fallopian tube | 0.82(0.30,2.21) | | 0.693 |
| Intestine | 3.58(1.60,8.02) | | 0.002 |
| Ovary | 1.37(1.10,1.70) | | 0.004 |
| Pelvic peritoneum | 1.11(0.89,1.38) | | 0.353 |
| Rectovaginal septum and vagina | 1.35(0.99,1.83) | | 0.057 |
| Uterus | 1.09(0.85,1.39) | | 0.492 |
| Unspecified | 1.18(0.87,1.60) | | 0.279 |

**Fig 3. Forest plots showing the causal estimates between LTL and endometriosis in different organs with IVW.** LTL, leukocyte telomere length; OR, odds ratio; IVW, inverse-variance weighted; CI, confidence interval.

male fertility potential [17]. TL has been found to be influenced by reactive oxygen species (ROS), with higher levels of ROS leading to shorter telomeres [18]. This is particularly relevant in the case of oligozoospermic individuals, where the level of ROS in spermatozoa is significantly elevated compared to fertile men [19]. Thus, infertile men have been found to have significantly shorter mean lengths of both LTL and STL [7]. And it can be concluded that shorter LTL is strongly associated with nonobstructive azoospermia (NOA) [20]. Nevertheless, upon meticulous exclusion of the confounding influences of age, BMI, and smoking, our study findings robustly challenge the prevailing belief that a causal link exists between LTL and male infertility or abnormal spermatogenesis. Importantly, these results align with a comprehensive investigation encompassing 599 individuals [21], lending further credibility to our conclusion. The discrepancy in findings between our study and Yang's study [20] may be attributed to differences in the definition of sperm abnormalities. Specifically, Yang's study found a significant relationship between LTL and NOA, whereas other types of sperm abnormalities were not significant. Furthermore, it is important to note that Darmishonnejad's study [7] did not account for the confounding effects of age, BMI, and smoking, and had a relatively small sample size, which increases the likelihood of false positive results.

Research suggests that various factors related to female infertility, including PCOS, diminished ovarian reserve (DOR), ovarian insufficiency, and tubal factor, have been linked to shorter telomere lengths in different cell types such as oocyte granulosa cells, endometrial tissue, and leukocytes [5]. However, there is controversy regarding the association between telomere length and female infertility, particularly in cases of PCOS [22,23] and Premature Ovarian Failure (POF) [24]. Our study aimed to investigate this association and found no causal relationship between telomere length and female infertility, regardless of the specific

factors involved. We hypothesize that the inconsistent findings in previous studies may be attributed to the influence of confounding factors. This highlights the major advantage of using MR analysis, which allows for the removal of confounding effects and provides more reliable results [25].

Endometriosis, a condition affecting approximately ten percent of reproductive-aged women, remains poorly understood in terms of its etiology and lacks curative treatments [26]. Current available treatments fail to effectively manage the associated symptoms, resulting in severe pain and decreased quality of life for women with endometriosis [27]. Despite the existence of screening tools, none can accurately identify or predict individuals or populations most susceptible to this disease, and no reliable biomarkers have been identified [28].

Women with endometriosis often exhibit characteristics such as high telomerase activity and higher human telomerase reverse transcriptase (hTERT) levels, which are linked to longer telomere lengths in eutopic secretory endometrial aberrations [29]. In contrast, the association between peripheral blood leukocyte telomere length and endometriosis has yielded conflicting results. Interestingly, a large-scale retrospective study that accounted for potential confounding factors such as age, body mass index (BMI), oral contraceptive use, and parity observed that shorter LTL was associated with a higher likelihood of a history of endometriosis [6]. Our findings align with this study, suggesting a potential role of LTL shortening in the pathogenesis of endometriosis. It is worth noting that ectopic endometriotic deposits, triggered by retrograde menstruation or gene activity, induce inflammation and cytokine release, leading to pro-proliferative changes in the eutopic endometrium and perpetuating the disease [30]. As peripheral blood leukocyte telomere length reflects the cumulative burden of inflammation [31], it is plausible that LTL shortening plays a role in the development of endometriosis. Furthermore, our results indicate a stronger causal association between LTL and deep abdominal locations such as the ovaries and rectum. Additional molecular experiments are necessary to confirm the relationship between leukocyte telomere length and endometriosis.

Understanding the causal association between telomere length and infertility could have far-reaching implications for both clinical practice and public health. Our results indicates the potential of telomere length as a biomarker for endometriosis is promising. Further research is needed to understand the underlying mechanisms and potential clinical applications of these findings. Additionally, studies with larger sample sizes and diverse populations would help to validate our results and strengthen the evidence for telomere length as a biomarker for endometriosis. Furthermore, elucidating the underlying mechanisms linking telomere length and endometriosis could pave the way for the development of novel therapeutic approaches targeting telomere maintenance and cellular aging.

However, this study does have certain limitations that should be acknowledged. Firstly, the sample size of infertility cases may not have been sufficient, potentially introducing bias into the study. Secondly, the data used in this study was limited to the European population, which may not accurately represent the global population. Therefore, it would be beneficial to expand the scope of the study to include a more diverse range of populations. Lastly, while statistical methods were employed to control for pleiotropy and heterogeneity, it is important to recognize that these factors may still have influenced the results.

## Conclusions

Our study utilized MR analysis with data summaries from a large sample GWAS analysis to investigate the causal association between LTL and the risk of male and female infertility, as well as abnormal spermatozoa in the European population. Our findings indicate that there is no significant causal relationship between LTL and these infertility factors. However, we did

observe a strong causal relationship between LTL and endometriosis, which has important implications for the diagnosis and prognosis of this condition.

## Supporting information

**S1 Fig. MR leave–one–out sensitivity analysis figure.**
(PDF)

**S1 Table. Summary of exposure and outcomes.**
(XLSX)

**S2 Table. Instrument variables.**
(XLSX)

**S3 Table. Summary of instrument variables.**
(XLSX)

**S4 Table. Odds ratios.**
(XLSX)

**S5 Table. Pleiotropy test.**
(XLSX)

**S6 Table. Heterogeneity test.**
(XLSX)

**S7 Table. Multiple Variants Mendelian Randomizaiton Result.**
(XLSX)

## Acknowledgments

The authors thank all participants and investigators for the contributions of GWAS data.

## Author Contributions

**Conceptualization:** Chenghui Zhao.

**Methodology:** Gaole An.

**Project administration:** Chenghui Zhao.

**Writing – original draft:** Gaole An, Xingnan Zhao.

**Writing – review & editing:** Chenghui Zhao.

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
