## [Decision Letter · Decision Letter 0]

11 Dec 2023

PONE-D-23-32653Unraveling the Causal Association Between Leukocyte Telomere Length and infertility: A Two-Sample Mendelian Randomization StudyPLOS ONE

Dear Dr. Zhao,

Thank you for submitting your manuscript to PLOS ONE. After careful consideration, we feel that it has merit but does not fully meet PLOS ONE’s publication criteria as it currently stands. Therefore, we invite you to submit a revised version of the manuscript that addresses the points raised during the review process.

The statistical analysis associated with MR analysis requires the confounding variates that were included in the studies.  The authors need to provide clarification of this issue. Also, please respond to each of the issues raised by the Reviewer. I look forward to your revision. Please submit your revised manuscript by Jan 25 2024 11:59PM. If you will need more time than this to complete your revisions, please reply to this message or contact the journal office at plosone@plos.org. Please include the following items when submitting your revised manuscript:A rebuttal letter that responds to each point raised by the academic editor and reviewer(s). You should upload this letter as a separate file labeled 'Response to Reviewers'.A marked-up copy of your manuscript that highlights changes made to the original version. You should upload this as a separate file labeled 'Revised Manuscript with Track Changes'.An unmarked version of your revised paper without tracked changes. You should upload this as a separate file labeled 'Manuscript'.If applicable, we recommend that you deposit your laboratory protocols in protocols.io to enhance the reproducibility of your results. Protocols.io assigns your protocol its own identifier (DOI) so that it can be cited independently in the future. For instructions see: https://journals.plos.org/plosone/s/submission-guidelines#loc-laboratory-protocols. Additionally, PLOS ONE offers an option for publishing peer-reviewed Lab Protocol articles, which describe protocols hosted on protocols.io. Read more information on sharing protocols at https://plos.org/protocols?utm_medium=editorial-email&utm_source=authorletters&utm_campaign=protocols.

We look forward to receiving your revised manuscript.

Kind regards,

Arthur J. Lustig, PhD

Academic Editor

PLOS ONE

Journal Requirements:

"NO authors have competing interests"

Reviewers' comments:

Reviewer's Responses to Questions

**Comments to the Author**

1. Is the manuscript technically sound, and do the data support the conclusions?

Reviewer #1: Yes

2. Has the statistical analysis been performed appropriately and rigorously? 

Reviewer #1: Yes

3. Have the authors made all data underlying the findings in their manuscript fully available?

Reviewer #1: Yes

4. Is the manuscript presented in an intelligible fashion and written in standard English?

Reviewer #1: Yes

5. Review Comments to the Author

Reviewer #1: In this investigation the author’s conduct a bidirectional MR study on the association between LTL and infertility. Overall, the report is concise and is careful to not overstate conclusions beyond the findings presented. However, key details of the statistical analysis and instrumentation are not well described and should be clarified in a revised submission. I suggest the following considerations:

MAJOR COMMENTS

• Did the authors perform adjustment for any important covariates in their analyses such as age and sex? Although MR operates under the assumption that instrumental variables do not act on confounders, it does not preclude the existence of those confounders. Important confounds of interest (age, sex, study site, BMI) should be included in the first stage regression to generate IVs for SNP-LTL and SNP-infertility related traits or as covariates in the second stage regression. For further details see Statistical Analysis of Kuo et al., 2019 (https://doi.org/10.1111/acel.13017) and Discussion of Burgess et al., 2011; https://doi.org/10.1093/ije/dyr036).

• Although the authors provide a supplementary table with a full list of SNPs utilized to generate IVs for LTL and infertility related traits, it would be beneficial to also include a summary table of the total number of SNPs associated with each factor and how much variance in each factor was explained by the overall instrument.

MINOR COMMENTS

• Results: it should be made clear that the data used for analyses of male and female infertility was derived from the FinnGen datasets in a similar manner as is described for the analyses on LTL and endometriosis.

• Line 164-168: It would help to contextualize contrasting results between TL and male infertility and their possible relationship to sample size by including the sample size of Ref #7 to contrast it with the “larger study” in Ref #20. That said, it is unclear if sample size contributes to the discrepancy of findings at all since the study on nonobstructive azoospermia (Ref #19) had a sample size of 866. Instead it might be a difference of outcome and how male infertility is defined.

6. PLOS authors have the option to publish the peer review history of their article (what does this mean?). If published, this will include your full peer review and any attached files.

Reviewer #1: **Yes: **Waylon J. Hastings

---

## [Author Response · Author response to Decision Letter 0]

11 Jan 2024

MAJOR COMMENTS

Reviewer #1: Did the authors perform adjustment for any important covariates in their analyses such as age and sex? Although MR operates under the assumption that instrumental variables do not act on confounders, it does not preclude the existence of those confounders. Important confounds of interest (age, sex, study site, BMI) should be included in the first stage regression to generate IVs for SNP-LTL and SNP-infertility related traits or as covariates in the second stage regression. For further details see Statistical Analysis of Kuo et al., 2019 (https://doi.org/10.1111/acel.13017) and Discussion of Burgess et al., 2011; https://doi.org/10.1093/ije/dyr036).

Response 1: Thanks for your valuable advice. We have thoroughly reviewed the recommended articles, specifically focusing on the statistical analysis conducted by Kuo. In their study, he used SNPs on mean LTL was previously estimated with adjustment for age, sex, body mass index (BMI), and smoking history (Haycock et al., 2017; https://doi.org/10.1001/jamaoncol.2016.5945). However, we were unable to obtain the same GWAS summary data used in their analysis. In our study, the summary GWAS data of LTL was adjusted for age, sex, array, and the first ten principal components. To account for the potential confounding effects of BMI and smoking history, we employed multivariable Mendelian randomization (MVMR) methods. The MVMR results have been included in Tables1-Table3, and further detailed information can be found in S7 Table. In summary, even after excluding the effects of confounding factors such as BMI and smoking history, we did not observe a significant causal relationship between LTL and infertility, while a statistically significant association between LTL and endometriosis was still observed. As suggested, we have provided a detailed description in the Exposure data section of the Methods and highlighted the changes in yellow.

Modified (LINE 75-83): The genetic association data for LTL were obtained from a genome-wide association study (GWAS) involving 472,174 European participants (10), adjusting for age, sex, and the first ten principal components (PCs). These participants were aged 40-69 years and had an equal distribution of males (45.8%) and females (54.2%). LTL measurements were determined using a quantitative polymerase chain reaction (qPCR) assay. To mitigate any potential biases stemming from body mass index (BMI) and smoking, we procured comprehensive BMI GWAS summary data from the UK Biobank and smoking GWAS sunmmary data from the GWAS & Sequencing Consortium of Alcohol and Nicotine use (GSCAN). Leveraging these datasets, we conducted multivariable Mendelian Randomization (MVMR) to ensure a robust analysis.

Reviewer #2: Although the authors provide a supplementary table with a full list of SNPs utilized to generate IVs for LTL and infertility related traits, it would be beneficial to also include a summary table of the total number of SNPs associated with each factor and how much variance in each factor was explained by the overall instrument.

Response 2: Thank you for your valuable suggestions. We acknowledge the significance of providing a comprehensive summary of the single nucleotide polymorphisms (SNPs) utilized as instrumental variables (IVs) for both telomere length (LTL) and infertility-related traits. To address this, we have included a summary table, namely S3 Table, in our revised manuscript. This table presents detailed information regarding the total number of SNPs associated with each factor, as well as the mean F-statistics and the sum of R2 values. In our study, we utilized a total of 135 instrumental variables. The mean F-statistics for these instrumental variables is approximately 120, indicating strong instrument strength. Furthermore, the sum of R2 values for the SNPs is approximately 0.01, suggesting that the SNPs explain a small proportion of the variance in the traits of interest.

MINOR COMMENTS

Reviewer #3: It should be made clear that the data used for analyses of male and female infertility was derived from the FinnGen datasets in a similar manner as is described for the analyses on LTL and endometriosis.

Response 3: Thanks for your valuable advice. We have added the data description in the results parts of male and female infertility.

Added (LINE 131-133): We conducted a comprehensive analysis using the male infertility and sperm abnormalities datasets from FinnGen to evaluate the causal association between leukocyte telomere length (LTL) and the occurrence of male infertility and sperm abnormalities.

Added (LINE 142-144): To investigate the potential causal relationship between leukocyte telomere length (LTL) and female infertility, we conducted a rigorous analysis utilizing the comprehensive collection of five datasets obtained from FinnGen. 

Reviewer #4: It would help to contextualize contrasting results between TL and male infertility and their possible relationship to sample size by including the sample size of Ref #7 to contrast it with the “larger study” in Ref #20. That said, it is unclear if sample size contributes to the discrepancy of findings at all since the study on nonobstructive azoospermia (Ref #19) had a sample size of 866. Instead it might be a difference of outcome and how male infertility is defined.

Response 4: We think this is an excellent suggestion. Upon further examination of the relevant literature, it was discovered that Reference #19 also supports the absence of a correlation between telomere length (LTL) and obstructive azoospermia (OA), while indicating a correlation between LTL and nonobstructive azoospermia (NOA). OA and NOA are two subtypes of oligozoospermia. Both Reference #7 and Reference #20 conducted research on male infertility populations, but they did not analyze the specific subtypes of oligozoospermia as explored in Reference #19. Reference #7 did not account for confounding factors such as age in their statistical analysis, whereas Reference #20 had a substantial sample size and effectively controlled for confounding factors including age and smoking, rendering its findings more reliable. Considering these factors, we have revised the discussion section regarding the association between LTL and male infertility.

Revised paragraph (LINE 200-209): Nevertheless, upon meticulous exclusion of the confounding influences of age, BMI, and smoking, our study findings robustly challenge the prevailing belief that a causal link exists between LTL and male infertility or abnormal spermatogenesis. Importantly, these results align with a comprehensive investigation encompassing 599 individuals [21], lending further credibility to our conclusion. The discrepancy in findings between our study and Yang's study [20] may be attributed to differences in the definition of sperm abnormalities. Specifically, Yang’s study found a significant relationship between LTL and NOA, whereas other types of sperm abnormalities were not significant. Furthermore, it is important to note that Darmishonnejad's study [7] did not account for the confounding effects of age, BMI, and smoking, and had a relatively small sample size, which increases the likelihood of false positive results.

---

## [Decision Letter · Decision Letter 1]

5 Feb 2024

Unraveling the causal association between leukocyte telomere length and infertility: a two-sample Mendelian randomization study

PONE-D-23-32653R1

Dear Dr. Zhao,

We’re pleased to inform you that your manuscript has been judged scientifically suitable for publication and will be formally accepted for publication once it meets all outstanding technical requirements.

Kind regards,

Arthur J. Lustig, PhD

Academic Editor

PLOS ONE

Additional Editor Comments (optional):

Reviewers' comments:

Reviewer's Responses to Questions

**Comments to the Author**

1. If the authors have adequately addressed your comments raised in a previous round of review and you feel that this manuscript is now acceptable for publication, you may indicate that here to bypass the “Comments to the Author” section, enter your conflict of interest statement in the “Confidential to Editor” section, and submit your "Accept" recommendation.

Reviewer #1: All comments have been addressed

2. Is the manuscript technically sound, and do the data support the conclusions?

Reviewer #1: Yes

3. Has the statistical analysis been performed appropriately and rigorously? 

Reviewer #1: Yes

4. Have the authors made all data underlying the findings in their manuscript fully available?

Reviewer #1: Yes

5. Is the manuscript presented in an intelligible fashion and written in standard English?

Reviewer #1: Yes

6. Review Comments to the Author

Reviewer #1: I appreciate the rigor and open-mindedness with which the authors approached their revision. I look forward to seeing this work provide further contribution to the literature on relationships between telomere length and infertility.

7. PLOS authors have the option to publish the peer review history of their article (what does this mean?). If published, this will include your full peer review and any attached files.

Reviewer #1: **Yes: **Waylon James Hastings

---

## [Editor Report · Acceptance letter]

13 Mar 2024

PONE-D-23-32653R1 

PLOS ONE

Dear Dr. Zhao, 

I'm pleased to inform you that your manuscript has been deemed suitable for publication in PLOS ONE. Congratulations! Your manuscript is now being handed over to our production team.

Kind regards, 

on behalf of

Dr. Arthur J. Lustig 

Academic Editor

PLOS ONE